# Regional Inequalities in Flood Insurance Affordability and Uptake under Climate Change

**Max Tesselaar [1],\*, W. J. Wouter Botzen [1,2,3], Toon Haer [1], Paul Hudson [4], Timothy Tiggeloven [1] and Jeroen C. J. H. Aerts [1,5]**

[1] Institute for Environmental Studies, Vrije Universiteit,
1081HV Amsterdam, The Netherlands; wouter.botzen@vu.nl (W.J.W.B.); toon.haer@vu.nl (T.H.);
timothy.tiggeloven@vu.nl (T.T.); jeroen.aerts@vu.nl (J.C.J.H.A.)

[2] Utrecht University School of Economics, Utrecht University, 3584EC Utrecht, The Netherlands

[3] Risk Management and Decision Processes Center, The Wharton School, University of Pennsylvania,
Philadelphia, PA 19104-5340, USA

[4] Institute of Environmental Science and Geography, University of Potsdam, 14476 Potsdam-Golm,
Germany; phudson@uni-potsdam.de

[5] Deltares, 2629 HV Delft, The Netherlands

\* Correspondence: m.tesselaar@vu.nl; Tel.: +31-205989557

**Abstract:** Flood insurance coverage can enhance financial resilience of households to changing flood risk caused by climate change. However, income inequalities imply that not all households can afford flood insurance. The uptake of flood insurance in voluntary markets may decline when flood risk increases as a result of climate change. This increase in flood risk may cause substantially higher risk-based insurance premiums, reduce the willingness to purchase flood insurance, and worsen problems with the unaffordability of coverage for low-income households. A socio-economic tipping-point can occur when the functioning of a formal flood insurance system is hampered by diminishing demand for coverage. In this study, we examine whether such a tipping-point can occur in Europe for current flood insurance systems under different trends in future flood risk caused by climate and socio-economic change. This analysis gives insights into regional inequalities concerning the ability to continue to use flood insurance as an instrument to adapt to changing flood risk. For this study, we adapt the "Dynamic Integrated Flood and Insurance" (DIFI) model by integrating new flood risk simulations in the model that enable examining impacts from various scenarios of climate and socio-economic change on flood insurance premiums and consumer demand. Our results show rising unaffordability and declining demand for flood insurance across scenarios towards 2080. Under a high climate change scenario, simulations show the occurrence of a socio-economic tipping-point in several regions, where insurance uptake almost disappears. A tipping-point and related inequalities in the ability to use flood insurance as an adaptation instrument can be mitigated by introducing reforms of flood insurance arrangements.

**Keywords:** climate change; flood risk management; insurance; socio-economic tipping-point; adaptation; partial equilibrium modeling

## 1. Introduction

Due to climate change, natural catastrophes such as flooding are likely to increase in frequency and severity in the future [1]. For 2019 alone, worldwide economic damage from flooding is estimated at $82 billion [2]. However, it is found that the impact of flooding disproportionally affects the world's poorer regions [3,4]. Furthermore, it is argued that the recovery process after a flood can reinforce

inequality, as more wealthy households can better access recovery mechanisms, e.g., insurance coverage [5], which allows them to get back on their feet quicker than those without. Flood insurance helps the insured to recover because it provides a contractual right for compensation after a flood. The resulting indemnity payments can enable a faster recovery, for example to repair damaged property, which may exceed the immediate available resources of a household [6,7]. Flood insurance can spread the risk of a large randomly occurring disaster over many households that are paying premiums, which makes a single household less sensitive to income shocks [8].

In addition to financing recovery, flood insurance systems can also be beneficial for limiting overall risk. For instance, by differentiating premiums according to the risk faced, a price signal of risk is generated. This price signal can encourage households to move away from—or avoid locating in—high-risk areas, or implement risk reduction measures such as dry- or wet-flood-proofing in exchange for premium discounts [6,9]. Risk-based premiums are therefore often considered to limit moral hazard, which is the behavioral response of insured individuals to limit effort to reduce risk. However, in the context of natural hazard insurance, evidence suggests that moral hazard is not apparent [10]. In private flood insurance systems, the premiums are often determined by the insurers themselves based on risk estimates from catastrophe models [11], like we use in our study for estimating local flood risk as well as for the risk reduction that can be obtained from flood-proofing measures by households. Alternatively, information from catastrophe models can be used by public authorities to regulate flood insurance premiums and premium discounts for risk reduction, as is common practice in public or public–private flood insurance systems. For example, in the National Flood Insurance Program in the US, premiums depend on flood risk zones and policyholders receive lower premiums when they elevate their home [12].

However, since risk-based premiums initially increase the cost of insurance in areas with high flood risk, the price signal may discourage households from obtaining insurance coverage. This process will be enhanced by climate change, and may lead to increasing social equity issues [7]. For example, in the US, it is found that risk-based premiums disproportionally impact low-income and minority populations [13], which can limit fair access to recovery mechanisms. Furthermore, the demand for insurance coverage can decline when premiums increase more rapidly than the perceptions of flood risk by households, which is likely to occur due to rising flood risk as a result of climate change. The result is a widening divergence between what households are willing to pay for insurance, based on their subjectively perceived threat of flooding, and what insurers must charge to remain in business [14]. This is a recognized issue regarding natural disaster insurance demand, because individuals tend to underestimate risk of events that occur with a low probability, but cause high damage, as is the case for flood risk [15].

Diminishing demand for flood insurance coverage can enlarge the burden of financing the recovery of flood damage from other sources, such as governmental compensation or private savings, which is especially problematic in the face of rising flood risk. These alternative forms of coverage are often considered inadequate, since for example government compensation, unless made explicit in law, leads to ambiguity regarding the compensation households receive for flood damage and results in unstable government budgets [16,17]. Besides this, private savings can be insufficient due to underestimation of flood risk by households, budget constraints [15], or competing objectives for how these savings should be used.

In a worst-case scenario, the aforementioned inequalities and the resulting lack of insurance uptake may result in a collapse of the flood insurance market in certain areas [18]. Such a development can also be interpreted as a climate induced socio-economic tipping-point, which is defined as a climate-change induced, abrupt change of a socio-economic system, into a new, fundamentally different state [19]. In this study, we examine impacts of the interaction between climate- and socio-economic change on the sustainability of flood insurance systems. Polhill et al. [20] note that while much attention has been placed upon tipping-points within ecological or economic systems independently, analyses of joint socio-environmental tipping-points are scarce, such as the impact of climate change on insurance systems. The relevance of this research topic is motivated by the observation that near collapses, or major reforms, of insurance systems have already been

observed in practice. For example, in the UK, where the autumn floods of 2000 marked the potential end of the 'universally available' flood insurance coverage [21]; or in Germany, where the floods of 2002 caused premiums to rise up to 50% and flood insurance penetration to decline by 10% to 20% in some areas [18]. Such sharp declines in insurance coverage have large negative welfare consequences, since impacts of uncovered flood damage have been associated with (mental) health problems [22], low regional economic growth rates [23], and social segregation of income groups [24].

The objective of this study is to identify areas where the functioning of (regional) private flood insurance markets could be hampered by rising unaffordability and declining demand for flood insurance, and to assess under which conditions of climate- and socio-economic change this occurs. We identify areas where the aforementioned problems with uncovered flood damage are the greatest and examine how policy adaptation can improve future resilience of flood insurance systems. Studied policy changes aim to improve affordability and insurance uptake, to secure the continued provision of flood insurance, and to suitably alter the financial burden upon households and governments. We assess measures ranging from insurance uptake requirements, introducing the government as a reinsurer, and a system with limited risk-based premiums where risk is shared amongst households to a certain degree [25].

Our study uses an adapted version of the "Dynamic Integrated Flood and Insurance" (DIFI) model, which is a partial economic equilibrium model that integrates flood risk, household behavior, and the insurance sector of different flood insurance systems for EU countries. We build upon the previous work in Hudson et al. [26] who examine desirable reforms of EU flood insurance systems based on national level DIFI output under a single future risk scenario, using a multi-criteria analysis. Here, we replace the flood risk assessment module in the previous DIFI model version with a more extensive and detailed flood risk dataset, which enables us to project regional inequalities in the impacts of climate change on existing flood insurance systems. While Hudson et al. [26] projected future flood risk under a high end climate change scenario combined with an average of socio-economic scenarios, the augmented DIFI model makes use of future risk projections under a variety of scenario combinations of climate and socio-economic change. This means that our approach allows for identifying specific future risk conditions under which the functioning of flood insurance systems is impaired. Moreover, we assess relevant DIFI model outcomes at a more spatially detailed regional level instead of the national level and present a more detailed analysis of the affordability and demand of flood insurance.

In the following section of the paper, the modelling approach will be explained, after which the results are presented, and a conclusion is given including policy recommendations.

## 2. Model Description

The DIFI model combines a riverine flood risk model with an insurance sector and household behavior model, in order to project future levels of unaffordability of premiums and insurance uptake under various existing and proposed flood insurance market structures in Europe. Each component of the model is described here in a separate section. Section 2.1 explains how flood risk is estimated; Section 2.2 describes how flood risk affects insurance premiums for various insurance schemes; and Section 2.3 defines the behavioral model by which unaffordability and uptake of insurance is determined. A list of variables and their definitions that are used in this section is provided in Appendix G.

This study uses an adapted version of the "Dynamic Integrated Flood Insurance" (DIFI) model developed by Hudson et al. [26]. The most notable change is the application of more detailed flood risk and socio-economic data, including multiple scenarios, which enables a more elaborate analysis of the occurrence of tipping-points for flood insurance in the EU. As we do not analyze the full spectrum of results produced by the DIFI model, we refer to the article by Hudson et al. [26] for a full model description. In this section, we specifically describe the process of obtaining unaffordability and market penetration projections that result from increasing flood risk under various flood insurance arrangements and climate change scenarios.

## 2.1. Flood Risk Model

In the flood risk model, the flood hazard (that is the probability of a flood of a certain magnitude occurring) is combined with the potential impact that can occur from flooding, which enables the estimation of flood risk, expressed as expected annual damage (EAD). EAD is important for the calculation of insurance premiums, as the insurance industry requires averaged expected damage values in order to spread flood risk over time using annual premium income. The data used to calculate the EAD in the current DIFI model version 2.0 are collected from the GLOFRIS model cascade [27,28], which is a change from DIFI model version 1.0, where data are obtained from the LISFLOOD model [29]. An advantage of this change is that the GLOFRIS model provides data using various combinations of climate and socio-economic scenarios, which enables the comparison of outcomes across a range of scenarios and controlling for uncertainty that underlies future projections of climate- and socio-economic change. An extensive validation exercise of the GLOFRIS model has been included in Ward et al. [27], which showed that the overall performance of the model is adequate and closely matches the LISFLOOD model.

The flood hazard is calculated in the GLOFRIS model for various representative concentration pathways (RCPs), which are scenarios for future levels of greenhouse concentrations. The RCPs are expressed as possible radiative forcing values in 2100, and existing trajectories include 2.6, 4.5, 6.0, and 8.5 W/m$^2$, respectively [30]. The model simulates water levels that occur with certain probabilities, which are expressed as return periods (2, 5, 10, 25, 50, 100, 250, 500, and 1000 years). For the baseline, the water level per river basin corresponding to each return period is determined by forcing the hydrological model PCR-GLOBWB [31,32] with meteorological fields (precipitation, temperature, radiation), using annual time-series data of maximum flood volumes for 1960–1999 from the EU-WATCH project [33]. For future projections (2030, 2050, and 2080) the model is run using bias-corrected meteorological data from the CMIP5 global circulation models (GCMs) [34], provided by the ISIMIP project [35].

The flood hazard is combined with exposure and vulnerability data in order to derive the potential impact it can cause [36]. Specifically, inundation maps are overlain with a land use map that identifies urban density, which for the baseline is taken from the HYDE database [37]. Current and future population data is taken from Huijstee et al. [38]. As flood insurance is usually organized on a household level, we estimate the number of exposed households by dividing the population by an average number of individuals per household on a country level, which is taken from Eurostat [39]. The economic value of exposed assets is dependent on national GDP per capita in 2010 [40]. Future simulations of built-up area are taken from Winsemius et al. [28], which uses data from the Shared Socio-economic Pathways (SSPs) [41], taken from IIASA's SSP database. Furthermore, the economic values of urban areas are determined by adjusting the GDP per capita of exposed population according to the SSP scenarios, which is also taken from the SSP database.

The potential damage a flood event can cause for each urbanized grid cell is then computed based on the extend of inundation, urban density, economic value, and global flood depth–damage functions per occupancy type taken from Huizinga et al. [42]. The global flood depth–damage functions account for the vulnerability aspect of the flood risk model and consist of stage–-damage curves, which project the impact of flood depth, and is based on the JRC global functions database. At this point, as is common practice, we take the average over the GCMs in order to limit biases originating from these models [43].

It is assumed that, in our modeling scheme, inundation only takes place when riverine water levels exceed the protection standards, i.e., when the protection standards of a dike are supposed to withstand riverine water levels of a 100-year return period, water levels of a return period lower than 100 years cannot lead to flooding. Regional protection standards are taken from the FLOPROS database [44], and are assumed to remain constant through time in regions where flood hazard increases. This means that, when flood hazard increases, investments are made in protection standards in order to account for rising water levels per return period, whereas in the case that flood hazard decreases, as a result of a dryer climate, the current protection standards will become higher in the future. In order to simulate the probability of the exceedance of protection standards, a damage

probability curve is fitted based on a power-law function. After this, the DIFI model uses a Monte Carlo approach in order to produce estimates of EAD and its variance, by drawing random return periods per region and comparing it to the current and future protection standards.

For further steps, the derived flood risk estimates are aggregated to the NUTS2 level, which is a classification of geographical regions in EU-countries and the UK. For certain countries, the NUTS2 regions comprise of provinces, such as in the Netherlands, Spain, and Italy, whereas for the Baltic states the NUTS2 regions make up the individual countries (Eurostat). Consistent with Hudson et al. [26], the flood risk module of the DIFI model focuses on households located in 1/100-year flood zones. These high-risk flood zones are most relevant for the focus of this study, since issues with affordability of premiums are most likely to occur, and the relevance of obtaining sufficient flood insurance coverage is highest in these areas. For example, in the US, the 1/100-year flood zones are designated by FEMA (Federal Emergency Management Agency) as "base flood" zones, and it is for households located in these regions that the flood insurance coverage debate is focused on most. For example, flood insurance coverage is mandatory for homeowners located in these regions with a mortgage from a federal lending institution [45]. While, traditionally in the UK, areas within the 1/75-year floodplain is the demarcation of universal insurance coverage.

Future climate events do not have an exact known probability distribution, which means that they can be characterized as being uncertain. We account for the uncertainty of future flood risk by assessing multiple scenarios of climate- and socio-economic change. In total, six RCP-SSP combinations are used, in order to examine a plausible range of potential future developments of flood risk. The "middle of the road" SSP2 scenario is assessed in combination with RCP2.6 and RCP6.0, in order to observe differences that arise from maintaining stringent greenhouse gas mitigation efforts (RCP2.6) versus a scenario that is judged to be realistic in the absence of mitigation efforts (RCP6.0).

Furthermore, scenario combinations RCP8.5-SSP3 and RCP2.6-SSP1 are assessed, which are included to compare extreme scenarios of climate risk development. The former combines the highest level of greenhouse gas emissions with an SSP scenario where mitigation and adaptation challenges are deemed difficult to meet due to regional conflicts and low technological advances [41], while RCP2.6-SSP1 has a strong emphasis on global sustainability.

Finally, RCP8.5-SSP5 is included in order to assess outcomes of differing socio-economic paths under a high climate-change scenario. In particular, SSP5 is considered a scenario representative of business-as-usual, with high socio-economic growth and a high societal dependence on fossil fuels [46]. We select RCP4.5-SSP2 in order to observe the impact of a scenario broadly aligned with the "Nationally Determined Contributions" (NDCs) of the Paris Climate Agreement [47]. The comparison of these scenarios provides insight into the potential consequences of failing to meet the climate change mitigation targets that underlie RCP4.5.

Baseline flood risk and projections of future EAD under the considered scenarios are shown in Appendix C. In Figure A3, it can be seen that riverine flood risk in 2010 is highest in southern Spain and northern Portugal, followed by regions in Poland, Croatia, Sweden, and the Mediterranean coast of France and Spain. The projected development of EAD until 2050 and 2080 are displayed in Figures A4 and A5 for the six scenarios considered in this study. Flood risk grows most rapidly under the high-end scenarios displayed in panels E and F, while not much difference can be observed between the remaining scenarios. Within the high-end climate change scenario RCP8.5, the highest impact on EAD is projected for the combination with SSP3.

### 2.2. Insurance Sector Model

The expected annual loss and volatility of losses are converted into insurance premiums for six stylized insurance market structures, which are derived in Hudson et al. [26] and partly based on Paudel et al. [48] and Paudel et al. [25]. Importantly, the current analysis is targeted at home insurance and disregards, for example, flood insurance for businesses or agriculture. The insurance market structures differ in terms of various market features such as purchase requirements, premium differentiation based on risk, and reinsurance arrangements. Selected European countries are

allocated to one of these stylized structures, based on their current flood insurance arrangement. Specifics concerning the included stylized insurance systems are presented in Table 1. The details of insurance arrangements for individual European countries differ from the stylized versions presented here. However, the most important characteristics that distinguish each system are captured by our model, which enables the comparison of the capacity of different insurance arrangements to deal with issues posed by climate change. For example, analyzed differences of flood insurance systems include risk-based premiums versus premiums that are unconnected to risk, voluntary versus mandatory insurance uptake, or a public versus a private reinsurance arrangement.

Each modelled insurance market structure has the same co-insurance style deductible of 15% of claimed loss, which is based on observed deductible levels reported in Paudel et al. [48]. Therefore, the expected annual loss transferred to the insurer is determined by taking the mean of 85% of the calculated loss for each region, and the volatility of losses is determined by the standard deviation of the transferred loss. The following section explains in detail how the premium is determined for the voluntary insurance system. It can be seen in Table 1 that this stylized insurance system is considered representative for most European countries. Furthermore, the map in Appendix D visualizes the allocated stylized insurance systems per country. As this system is also most important for the current analysis on insurance tipping-points, due to its risk-based premiums and voluntary insurance uptake, we discuss its premium structure here and refer to Appendix A for a description of the premium structures for the remaining insurance systems. We do not explicitly model insolvency of insurers, but we do account for the practice to minimize this risk through reinsurance. Within the EU there is strict supervision regarding the solvency of insurers covering natural or man-made disasters (see [49] for an overview of this regulatory oversight from the EU). As a result, presently the risk of bankruptcy of individual insurers is unlikely. In particular, the annual insolvency probability should be below 0.5% to comply with the Solvency II European Union legislation.

The text below is based upon that presented in Hudson et al. [26] and re-presented here in order to briefly describe the model used, of which more details can be found in Hudson et al. [26].

**Table 1.** A summary of the stylized flood insurance arrangements to which European countries are allocated.

| Structure Group | Sector Covering Flood Risk | Common Market Features | Countries Allocated |
|---|---|---|---|
| M1. Solidarity public structure | Public | - Mandated purchase requirement<br>- Premiums unconnected to risk<br>- Very high penetration rate (100%)<br>- Government support for extreme losses | France; Belgium; Spain; Romania |
| M2. Semi-voluntary private market | Private | - Purchase is connected to mortgage lender conditions<br>- Premiums are risk-based<br>- High penetration rate (75–100%)<br>- Damage to buildings is more often insured than contents due to mortgage requirements<br>- No government support for extreme losses | Sweden; Ireland; Hungary; Finland |
| M3. Voluntary private market | Private | - No government mandated purchase requirement (voluntary)<br>- Premiums are risk-based<br>- Medium to low penetration rates (25–60%) if government support is uncertain (e.g., Germany)<br>- Very low penetration rates (0–25%) if government support is certain (e.g., Austria)<br>- Possible government reinsurer rather than government compensation | Austria; Netherlands; Germany; Italy; Portugal; Luxembourg; Greece; Poland; Czech Republic; Slovakia; Slovenia; Croatia; Bulgaria; Latvia; Estonia; Lithuania |
| M4. Voluntary PPP market | Public-Private | - Voluntary market<br>- Government reinsurer for extreme risk<br>- Other characteristics are the same as M3 | Hypothetical market structure |
| M5. Semi-voluntary PPP market | Public-Private | - Semi-voluntary market.<br>- Government reinsurer for extreme risk<br>- Other characteristics are the same as M2 | Hypothetical market structure |
| M6. Public-Private Partnership (PPP) market | Public-Private | - Purchase is connected to mortgage lender conditions.<br>- Premiums are partially connected to risk<br>- High degree of loss sharing<br>- High penetration rate (75–100%)<br>- Damage to buildings is more often insured than contents (due to mortgage requirements)<br>- Government reinsurer rather than government compensation | UK |

Notes: Romania displays many of the criteria of M1 with the exception of the insurance penetration rate being less than 20% due to poor enforcement of the purchase requirements. Therefore, it has been placed within M1 as a stylized market structure. *Source:* Hudson et al. [26].

Voluntary or Semi-Voluntary Market (M2, M3)

Equation (1) shows how the premium is determined for the (semi-)voluntary insurance systems, which follows a flood insurance pricing rule used in various studies [48,50,51]. In this system, there is a full risk signal from insurers to policyholders. The premium is determined at the NUTS2-level hence the subscript $j$ is introduced, which represents that geographic level. This implies that expected flood damage ($\bar{L}_{j,t}$) and the deductible ($\bar{D}_{j,t}$) are modelled per high-risk household on the NUTS2-level at time $t$.

Specific cost loading factors are introduced for the proportion of risk that is covered by the private primary insurer, or the private reinsurer. $\dot{\lambda}_{c,t}$ is the cost-loading factor for the primary insurer, which is implemented on country-level as indicated by subscript $c$; $\ddot{\lambda}_{c,t}$ is the cost-loading factor for the reinsurer; the superscript $RR$ indicates the risk transferred from the insurer to the reinsurer; $r$, is the risk aversion coefficient of the primary insurers and reinsurers; $\sigma_{0<a<99.8}$, is the volatility of flood damage within a quantile range that is considered insurable [25].

The loading factors for the primary insurer and reinsurer are determined as follows. For the private primary insurers Bertrand competition is assumed, which means that no company can charge a profit-loading factor. The cost-loading factor for private reinsurers is fixed at $\ddot{\lambda}_{c,t} = 0.5$, as the market tends to be more concentrated [26]. The cost-loading factors are determined on a country-level based on OECD insurance statistics of costs of non-life insurance policies, as is described in detail in Hudson et al. [26].

$$
\begin{aligned}
\bar{\pi}_{j,t} = \left(1 + \dot{\lambda}_{c,t}\right)&\left(E\left(\bar{L}_{j,t}(p) - \bar{D}_{j,t}(p)\right) + r * \sigma_{0<a<99.8}\right) \\
&+ \left(1 + \ddot{\lambda}_{c,t}\right)\left(E\left(\bar{L}_{j,t}^{RR}(p) - \bar{D}_{j,t}^{RR}(p)\right) + r * \sigma_{0<a<99.8}^{RR}\right)
\end{aligned}
\tag{1}
$$

The final premiums faced by individuals ($\pi_{i,j,t,}$) is determined as the baseline average risk per household (indicated by subscript $i$) for a certain NUTS2 region, which is reduced by the premium discount ($ER_{DRR}$) that is determined by the implementation of risk reduction measures (Equation (2)).

$$
\pi_{i,j,t} = (1 - ER_{DRR})\bar{\pi}_{j,t}
\tag{2}
$$

*2.3. Household Behavior Model*

The estimated premiums for the included insurance market structures are then fed into a household behavior model, which projects flood insurance demand, and allows us to derive the level of unaffordability and market penetration, which are the main interest for this study. Moreover, the household behavior module of the DIFI model also involves household decisions to apply risk reduction measures (see Hudson et al., [26] for details).

The behavioral model representing households at risk of flooding is based on subjective expected utility theory [52]. The decision to obtain insurance coverage is only made when uptake is voluntary. Each household included in the model purchases insurance if that option yields the highest expected utility, or chooses to remain uninsured if that expected utility is higher instead, as represented by the two elements in Equation (3). The first element ($U_{1,i,j,t,s}$) expresses the expected utility of not being insured, in which case the household is faced with uncompensated flood losses with probability $p$. The second element ($U_{2,i,j,t,s}$) expresses the expected utility of being insured, where the household pays the premium and faces the deductible in the case of flood damage. New terms in Equation (3) includes $W_{i,j,t}$, which symbolizes estimated wealth of household $i$ in region $j$ at time $t$. Wealth is determined as a fixed ratio of income, which is estimated by taking households' stock of financial assets as a percentage of disposable income, as indicated by Eurostat [53]. $\alpha_i$ represents the risk aversion of a household and is fixed at $\alpha_i = 1$, resulting in a logarithmic utility function. Subjective flood occurrence perceptions are included by modelling households to consider flood losses over the probability range $[0, \widehat{PS}_{i,j}]$, where $\widehat{PS}_{i,j} = \vartheta_{i,j} PS_j$ where $PS_j$ is the regional flood protection standard and with $\vartheta_{i,j}$ being a distribution of subjective flood occurrence probabilities (see Hudson et al. [26] for a detailed discussion on how this is done). Therefore, households'

subjective flood probabilities differ from the objective probability determined in the flood risk module. $\gamma_{i,j}$ is a parameter between 0 and 1, which represents how the perceived flood impact differs from the modelled flood impact, which is similarly taken from a generalized Pareto distribution, calibrated using data about flood insurance uptake from the German Insurance Association [54].

$$E(U) = \begin{cases} E(U_{1,i,j,t,s}) = \int_0^{p=\widehat{PS}_{i,j}} p\,ln\big(W_{i,j,t} - \gamma_{i,j}L_{i,j,t}(p)\big)\,dp \\ E(U_{2,i,j,t,s}) = \int_0^{p=\widehat{PS}_{i,j}} p\,ln\big(W_{i,j,t} - 0.15\gamma_{i,j}L_{i,j,t}(p) - \pi_{i,j,t,s}\big)dp \end{cases} \tag{3}$$

The household, moreover, faces a budget constraint, where it is assumed that no insurance is purchased when the premium is deemed unaffordable, as shown in Equation (4). This budget constraint follows the theory of hierarchy of human needs [55], where flood insurance may be considered subordinate to primary economic goods such as food and shelter. This is conceptually similar to Bundorf and Pauly [56], who argue that insurance affordability can be judged as having sufficient income to afford a certain standard of living after purchasing insurance. Unaffordability is measured following Hudson et al. [51], where insurance is considered unaffordable when the sum to be paid causes the household's poverty-adjusted disposable income ($Income_{i,j,t}$) to descent below the poverty line ($Poverty\ Line_{c,t}$), which is set at 60% of national median income. The magnitude of unaffordability is measured as the regional total value of the unaffordable share of the premium for each household.

Future unaffordability of premiums is dependent on the level of income growth households may experience. In the DIFI model, household income growth is modelled to follow GDP-growth on the NUTS2-level according to the respective SSP scenarios. Therefore, GDP-growth is considered to increase exposure and flood risk, while it is also modelled to increase household income and limit the unaffordability of premiums. For every modelled household, income is determined by randomly drawing from a log-normal income distribution. Although the position of the distribution is calibrated per country and changes over time depending on the SSP-projections, the shape is applied equally for EU countries. This is done because the development of the shape of income distributions is uncertain and is affected by future income redistribution policies, such as through the income tax system and social safety nets.

Equation (4) summarizes the modelled decision framework. Households will buy insurance if it maximizes expected utility, as long as it is affordable to do so.

$$U = \begin{cases} insure & if & E(U)_{1,i,j,t,s} < E(U)_{2,i,j,t,s} & s.t.\,\pi_{i,j,t,s} \leq Income_{i,j,t} - Poverty\ Line_{c,t} \\ not\ insure & if & E(U)_{1,i,j,t,s} \geq E(U)_{2,i,j,t,s} & or\ \pi_{i,j,t,s} > Income_{i,j,t} - Poverty\ Line_{c,t} \end{cases} \tag{4}$$

We realize that affordability is a normative concept [57], which results in different metrics being proposed for measuring it. Hudson [58] observes three main affordability metrics regarding flood insurance: Residual income—the purchaser must be left with a certain income after the purchase [59]; expenditure cap—the purchaser must not spend more than a given percentage of income [7]; housing burden—the combined expenditure on housing must not exceed a certain percentage of income [45]. Given the current study's focus at the European scale, measuring unaffordability following the residual income approach used in Hudson et al. [51] is most suitable. This is because the at-risk-of-poverty line employed is a pre-existing policy relevant definition across Europe for measuring financial burden [60]. On the other hand, determining suitable income thresholds for the expenditure and housing burden definitions would require detailed stakeholder consultation within each country studied. Moreover, the residual income approach directly accounts for equity concerns, which is important given the increasing role households play in integrated flood risk management, as noted in international agreements such as the Sendai Framework for Disaster Risk Reduction [61]. Equity is accounted for by considering a household's position in the overall income distribution, as well as their available resources.

An elaborate sensitivity analysis of the behavioral module in the DIFI model is performed in Hudson et al. [26], which includes several alternative assumptions concerning the utility function

determining insurance demand. For example, instead of expected utility theory, the model is operated using a household decision function based on prospect theory. Moreover, the sensitivity analysis includes the use of various levels of risk aversion and flood risk perceptions. Overall, the changes in insurance demand caused by the alternative decision theory or behavioral assumptions is limited. Considering the scope of the current study we decided to maintain the utility theory based on subjective expected utility. Instead of focusing on how alternative decision theories or behavioral parameters impact the occurrence of an insurance tipping-point, the main focus of the current study is to analyze the impact of different scenarios of climate and socio-economic change.

## 3. Results

### 3.1. Unaffordability of Flood Insurance under Current Insurance Arrangements

This section presents the projections of unaffordability of flood insurance premiums under the current stylized insurance arrangements (Table 1). The figures included in the text show results for the low-end scenario (RCP2.6-SSP1), the "Paris climate accord scenario" (RCP4.5-SSP2), as well as a high-end scenario (RCP8.5-SSP3), because these scenarios capture the lower and upper bound of outcomes. The remaining scenarios are included in Appendix B.

Before we show the development of unaffordability of flood insurance premiums that results from climate- and socio-economic change, it is useful to consider the baseline premiums and the percentage of the population for whom these are considered unaffordable. This is shown in Figure 1, for current national flood insurance arrangements, as shown in Table 1. The highest premiums are estimated for high-risk areas in Portugal and Poland, followed by several regions in Germany, Croatia, and the Baltic states. High levels of unaffordability are consequently seen in Portugal, Poland, Bulgaria, and the Baltic states, amongst several other regions. Premiums can become unaffordable for a large share of the population in areas at high risk of flooding when insurance premiums are fully risk-based, and when household income is low.

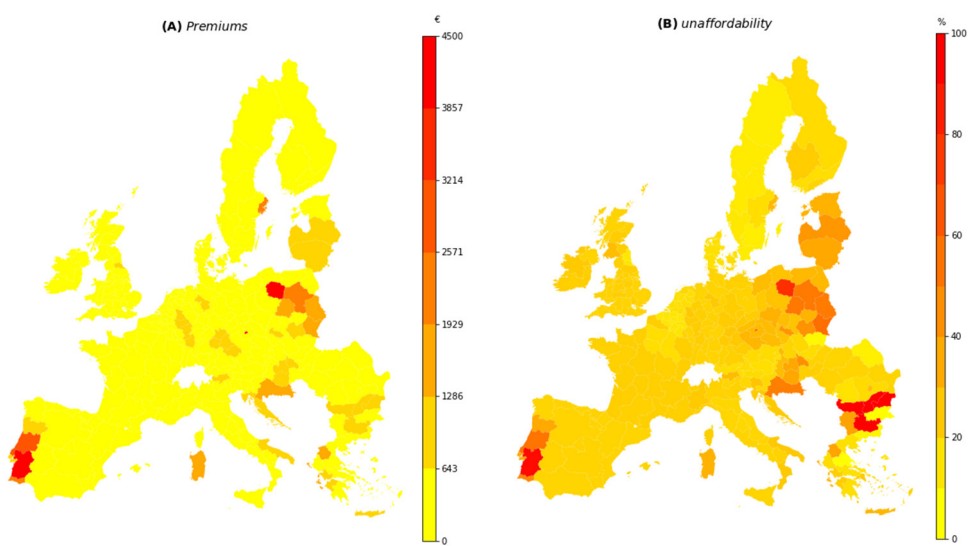

**Figure 1.** Flood insurance premiums in high-risk areas per NUTS2-region in 2010 under status-quo flood insurance arrangements (**A**) and the percentage of the population for whom premiums are deemed unaffordable in 2010 (**B**).

Figure 2 shows the percentage change of the share of the population that cannot afford flood insurance, up to 2050 and 2080, while keeping the current market arrangements constant for each country. As can be seen from panels A and B corresponding to RCP2.6-SSP1, high growth rates of unaffordability are a continuation from the current status in Figure 1, and are mostly found in Eastern European countries, in addition to several regions in Sweden, Portugal, and Italy. General reasons

for higher growth of unaffordability in these regions are more severe increases in flood risk, that, depending on the insurance market structure, translates into higher premiums (see Appendixes C and E for projected EAD and premiums under the scenarios plotted in Figure 2). The combination of this with below-average projected income growth for most of these regions, leads to a relatively larger increase in unaffordability of flood insurance than in many Western European regions.

The results for RCP4.5-SSP2 (panels C and D) and RCP8.5-SSP3 (panels E and F) show a consistent increase in the inequality of insurance unaffordability projections between regions compared to panels A and B. This effect can be explained by steadily worsening flood risk, as well as declining economic growth and increasing income inequality over the shown scenarios. The sharp increase in the projected unaffordability of flood insurance premiums after 2050 is roughly aligned with the increase in flood risk in Europe for that period found by Alfieri et al. [62], as well as the expected increase in exposure to flooding due to socio-economic development.

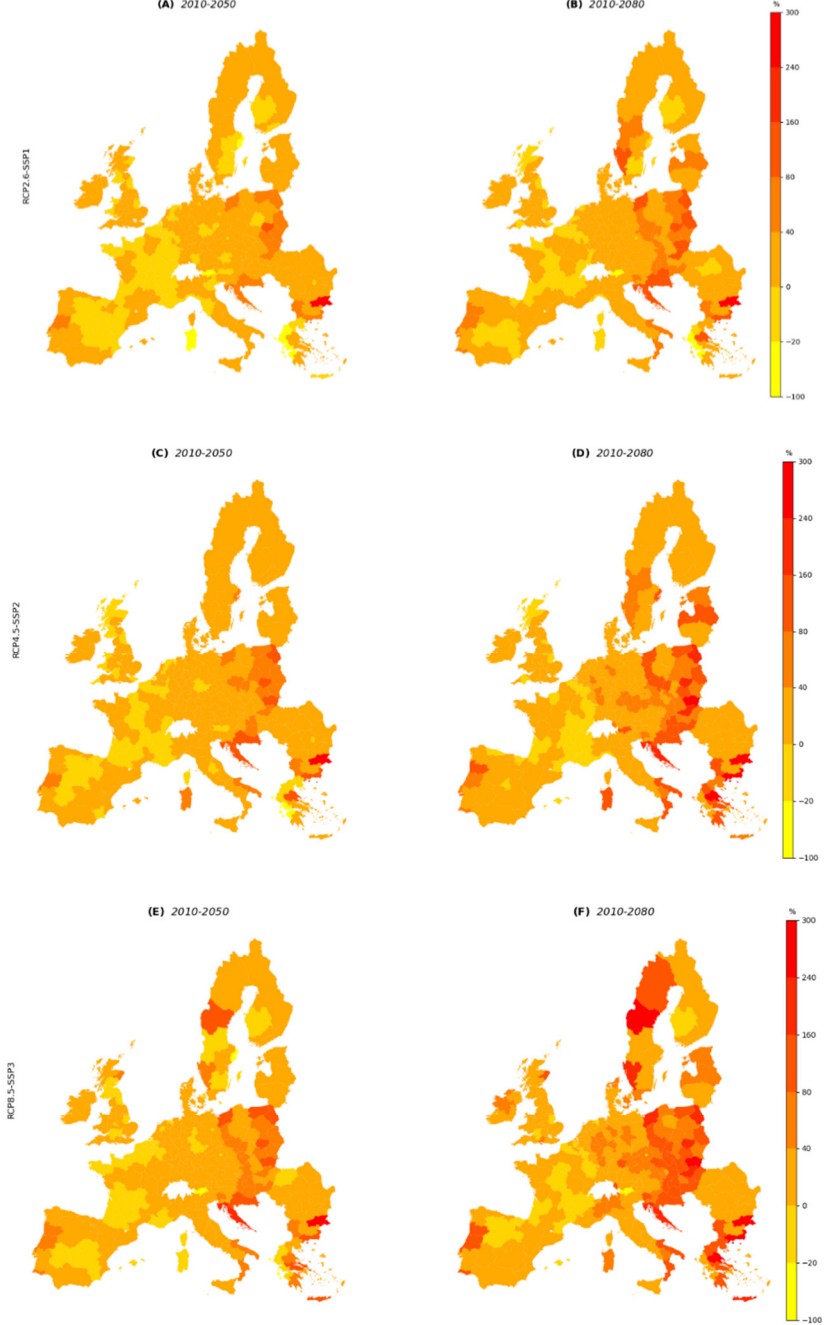

**Figure 2.** The percentage change of unaffordability under status-quo insurance arrangements for households in high-risk areas under the RCP2.6-SSP1 (panels **A** and **B**), RCP4.5-SSP2 (panels **C** and

**D**), and RCP8.5-SSP3 (panels **E** and **F**) scenarios of climate- and socio-economic change for the periods 2010–2050 (left) and 2010–2080 (right). Unaffordability is measured as the percentage of the population in high-risk areas that cannot afford the insurance premium.

Particularly in the RCP8.5-SSP3 scenario, it is visible that many regions are expected to experience a sharp increase in the unaffordability of insurance. Until 2080, there are 26 NUTS2 regions where the population for whom flood insurance is unaffordable is projected to more than double, and where in 2010, premiums were already unaffordable for more than 10% of the population in high-risk areas. For voluntary, risk-based, insurance structures, a high rise in unaffordability triggers lower insurance uptake, as affordability is a condition for the uptake of insurance in our stylized voluntary insurance system. Countries with a solidarity market structure (Spain, France, Belgium, and Romania) are more effective at limiting the growth of unaffordability, as most of the risk in the high-risk areas that are studied here is transferred to regions with lower risk. However, with modeled insurance premiums being unaffordable for 18% of the population in France in 2010, this insurance system requires additional financial assistance to prevent unaffordability for low-income households, even though the magnitude of unaffordability burdens are relatively low.

Comparing the different scenario projections for the unaffordability of flood insurance (Figure 2 above and Figure A1 in the Appendix B), consistency can be found across scenarios in terms of the regions that are most impacted by the expected rise in unaffordability. Whereas it is clear that under RCP2.6-SSP1 the rise in unaffordability is lowest for most regions, for the remaining scenarios the differences are subtle. This outcome supports the robustness of our results to variability in climate- and socio-economic change, and substantiates the need for policy change in order to avoid this development.

## 3.2. Insurance Market Penetration

The previous section showed how increasing flood risk and expected future income for households can lead to higher unaffordability of flood insurance premiums. In countries with a voluntary insurance market, higher premiums can cause a decline in demand for flood insurance, which, in extreme cases, can cause flood insurance to disappear in high-risk areas.

This section examines the expected development of flood insurance demand by households in future flood risk scenarios. Figure 3, below, shows the outcomes of insurance penetration as a percentage of the population in areas at high risk of flooding for the current market structures. Countries with a solidarity market structure are shown to have a penetration rate of 100% as uptake of insurance is mandatory, and those countries that have a semi-voluntary market structure are shown to have a penetration rate of 75%, which approximates penetration rates of building insurance for these countries as shown by literature [63].For most countries displayed in Figure 3, insurance is voluntary, resulting in lower penetration rates. The reason for this is that, although increasing flood risk raises the expected utility of insuring, the insurance premium can rise relatively more than the expected flood risk as a result of the loading factor charged by private insurers and reinsurers, which ultimately reduces the incentive to insure. When premiums exceed the willingness to pay for flood insurance, which is dependent on subjective risk perceptions, or when spendable income is insufficient, households will refrain from purchasing flood insurance when this is optional.

For certain regions in Portugal, Bulgaria, Poland, and the Netherlands, almost none of the households in high-risk areas obtain insurance coverage in 2010. In other regions with voluntary insurance, penetration rates range between 20% and 60%. The projection of insurance uptake for 2050 and 2080 under RCP2.6-SSP1 shows a slight decline (panels A–C), which declines progressively further under scenarios RCP4.5-SSP2 (panels D–F) and RCP8.5-SSP3 (panels G–I). The largest decline is observed under RCP8.5-SSP3 as this scenario combines the highest increase of flood hazard with the lowest economic growth rates, which causes insurance premiums to make up a larger share of spendable income, discouraging the uptake of flood insurance.

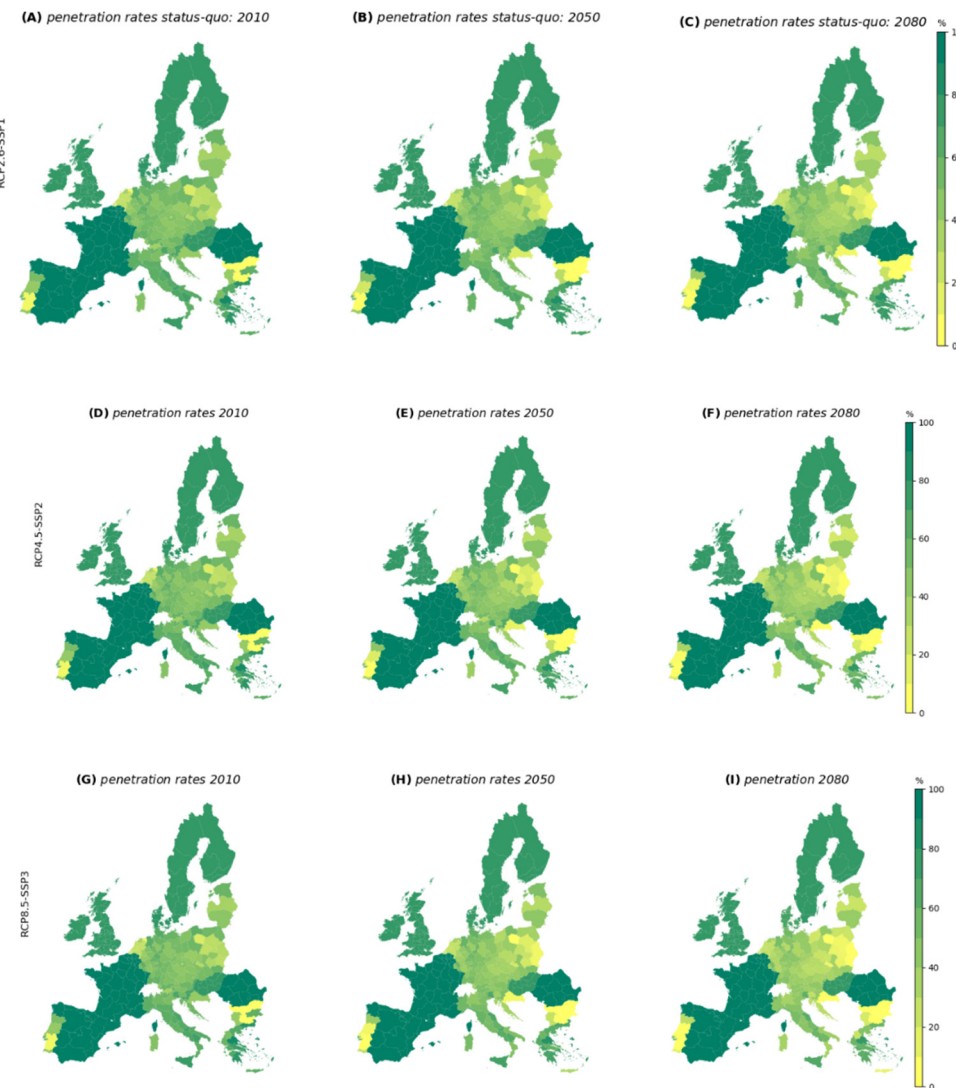

**Figure 3.** Flood insurance penetration rates under status-quo insurance arrangements for households in high-risk areas under the RCP2.6-SSP1 (Figure 3 **A–C**), RCP4.5-SSP2 (Figure 3    **D–F**), and RCP8.5-SSP3 (Figure 3    **G–I**) scenarios of climate and socio-economic change for the periods 2010 (left), 2050 (middle), and 2080 (right).

In Figure 3, it can be seen that several regions are expected to experience low penetration rates by 2080. Regions where an insurance tipping-point is projected, that is, where demand for flood insurance reduces to such an extent that it almost disappears, are presented in Table 2. The table shows the country names followed by the share of the total amount of NUTS2 regions where this decline in insurance demand is expected, per RCP-SSP combination. The presented regions initially have a penetration rate above 20%, which, by 2080, is projected to decline to below 10%. In Table 2, it can be seen that, under RCP8.5-SSP3, 11 regions are projected to experience a tipping-point under these criteria. More stringent criteria, such as initial penetration rates above 30%, decreasing to less than 5% in 2080, changes this result to three regions. The criteria used here are meant to illustrate our definition of a socio-economic tipping-point, and are not meant to represent a threshold used in the literature. In the tipping-point regions, the private flood insurance market in high-risk areas is expected to (almost) disappear due to low demand for coverage, as a result of increasing flood risk. Therefore, a significant change in the flood insurance sector can be expected for these regions, where instead of a formal insurance arrangement, governments or households themselves may bear the responsibility of covering the damage ex post.

Unsurprisingly, RCP8.5-SSP3 shows the highest number of tipping-point regions, as the high premium increase and low economic growth under this scenario is already shown to lead to the increase of unaffordability of insurance compared to other scenarios. Contrary to this, RCP2.6-SSP1 shows the lowest number of tipping-point regions. Across scenarios, the variability of the number of regions where insurance uptake is expected to diminish is moderate, ranging from 5 (RCP2.6-SSP1) to 11 (RCP8.5-SSP3), which expresses the severity of the issue. Considering the countries that appear in Table 2, it can be seen that certain Eastern European regions appear disproportionally often compared to Western European regions. This can be explained by a combination of a high rise in riverine flood risk in these regions [62], as well as economic growth rates that are below European average for these regions.

The previous results are modeling outcomes where the status-quo of flood insurance is assumed to remain unchanged per country. In the following section, we show how certain policy changes can improve the expected future performance of flood insurance, both in terms of unaffordability and insurance uptake. We will focus our analysis specifically on the regions where a tipping-point is projected, which are the regions where the impact of climate- and socio-economic change on flood insurance is the greatest.

**Table 2.** Regions where a tipping-point in flood insurance occurs by 2080, per scenario. The number of regions is the first number in brackets after the country name. The second number is the total count of NUTS2 regions per country.

| RCP4.5-SSP2 | RCP8.5-SSP3 | RCP2.6-SSP1 | RCP8.5-SSP5 | RCP6-SSP2 | RCP2.6-SSP2 |
|---|---|---|---|---|---|
| Croatia (1/2) | Croatia (1/2) | Croatia (1/2) | Croatia (1/2) | Croatia (1/2) | Croatia (1/2) |
| Bulgaria (1/6) | Bulgaria (1/6) | Bulgaria (1/6) | Bulgaria (1/6) | Bulgaria (1/6) | Bulgaria (1/6) |
| Poland (6/16) | Poland (6/16) | Poland (2/16) | Poland (6/16) | Poland (5/16) | Poland (5/16) |
| Portugal (1/5) | Portugal (1/5) | Portugal (1/5) | Portugal (1/5) | Portugal (1/5) | Portugal (1/5) |
| Czech (1/8) | Czech (1/8) | | | | |
| | Greece (1/13) | | | | |

### 3.3. Unaffordability of Flood Insurance and Penetration Rates under Policy Change

In order for private flood insurance to remain operational under future scenarios of high flood-risk and low income growth for households, it may be important to implement changes to national flood insurance systems. In this section, we show how the previously stated issues with flood insurance can be limited by means of several policy changes as proposed in Paudel et al. [25] and Hudson et al. [26].

A change to a voluntary PPP market (M4) replaces the private with a public reinsurer, which lowers the reinsurance costs faced by the primary insurer. This policy measure results in improved insurance penetration rates ranging between 0 and 10 percentage points by 2080, as can be seen in Figure 4.

After introducing a public reinsurer, the next policy measure in our analysis is to make flood insurance uptake semi-voluntary (M5), for instance by making coverage a prerequisite for obtaining a mortgage. This results in flood insurance penetration rates of higher than 75%, as empirical evidence suggests [63]. The level of unaffordability is identical to the voluntary-PPP market structure, as the premiums are not affected by higher insurance uptake in our analysis. Nevertheless, by expanding insurance uptake within high-risk areas, the population there is financially better protected against flood damage. Moreover, forcing higher insurance uptake, in combination with risk-based premiums, can enhance risk awareness and can stimulate risk-reduction measures if the premiums are sufficiently responsive to policyholder risk reduction activity [64] and do not lower the immediate financial capacity of policyholders to afford such measures.

The final policy measure in our analysis is to introduce a stronger risk-sharing mechanism, or premium cap, that prevents premiums from rising excessively. The premium cap used in this study is based on that applied by Flood Re in the UK, which maintains a maximum premium roughly equal to 1.8% of national median income [26]. The difference between the risk-based premium and the

actual premium is then transferred to low-risk households. Figure 5 presents the reduction in unaffordability that results from this policy measure compared to status-quo insurance arrangements in 2080. It can be seen that the percentage of the population that cannot afford premiums in 2080 is significantly lower than in the voluntary market. More specifically, the reduction in unaffordability ranges between 60% and 100%, with the largest improvements seen in regions in Portugal, Bulgaria, and Croatia. As this study specifically analyzes the households facing high risk, it is sensible that risk-based premiums are reduced considerably after introducing a stronger risk-sharing mechanism amongst the total population. For this reason, unaffordability is reduced by almost 100% for some regions.

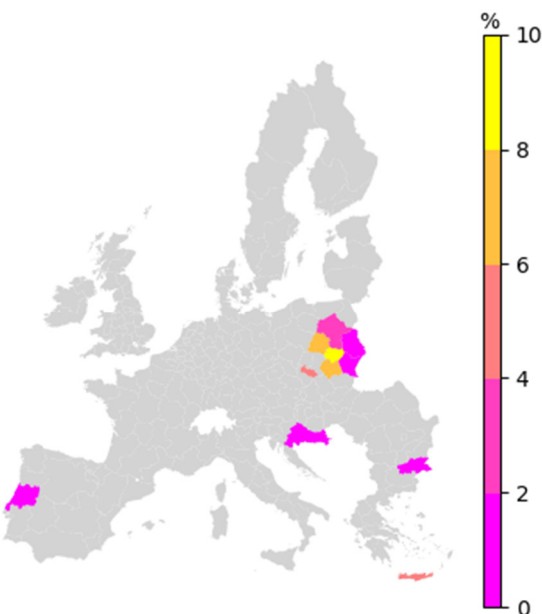

**Figure 4.** The reduction of unaffordability as a result of replacing a private with a public reinsurer.

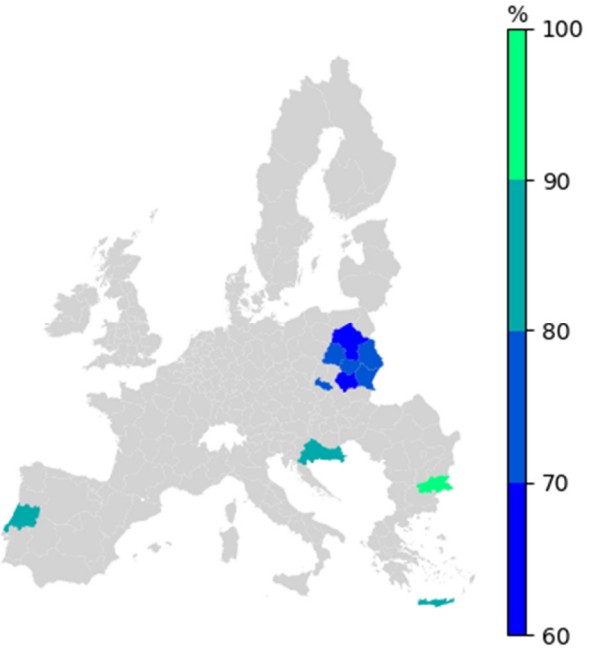

**Figure 5.** The reduction of unaffordability as a result of replacing the private voluntary with a PPP-insurance market.

A deliberation for the proposed policy measures is that a degree of unaffordability of premiums still persists, which can be problematic when maintaining mandatory insurance uptake. Therefore, supplementary policies, such as a subsidy or voucher, are required to alleviate the unaffordable share of the premium. For this reason, it is important to consider the total value of unaffordability, besides the share of the population that cannot afford flood insurance. This is presented in Figure 6 for the PPP-insurance market in 2080 under RCP8.5-SSP3. In the following section, we will elaborate on the policy design of such a subsidy or voucher system.

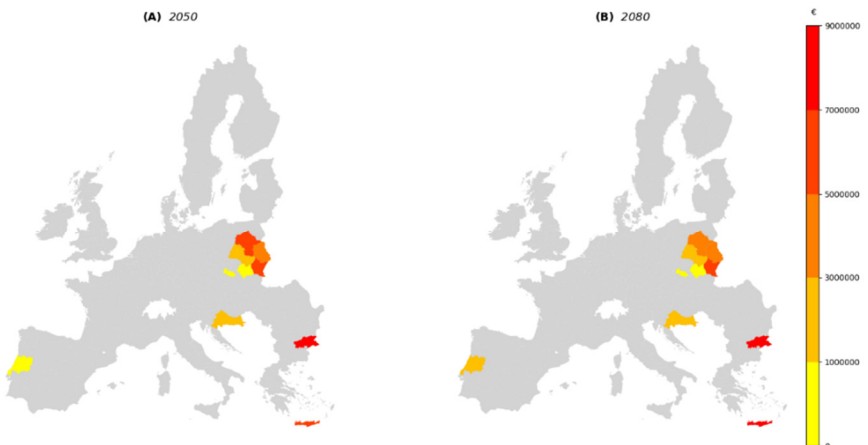

**Figure 6.** The total value of unaffordability under the PPP-insurance market, in Euros (2013), for the selected regions where a tipping-point is expected.

## 4. Discussion

This analysis uses an adapted version of the DIFI model (V2.0), in which flood damage- and socioeconomic data input is acquired from the GLOFRIS framework [27,28]. This model adjustment facilitates running the model using various flood hazard and exposure data projections, which are incorporated as RCP-SSP scenario combinations. Previous studies highlight the importance of comparing various risk projections, because these can create large differences in outcomes [41], meaning that optimal adaptation strategies can differ based on the future scenarios [27]. For example, Arnell and Lloyd-Hughes [65] find that the consequences of floods under the same RCP scenario diverge significantly between levels of socio-economic development, with higher adaptation challenges leading to higher impacts. Similarly, we find notable differences between SSP scenarios when holding the RCP scenario constant. The impact of socio-economic development becomes greater for high climate change projections (RCP8.5) compared to low climate change projections (RCP2.6), due to a higher average frequency and intensity of floods in the former [66].

Whereas the analysis in the current study is sensitive to various future flood risk scenarios, as well as multiple types of flood insurance markets, the household behavioral module based on subjective expected utility theory is the same between households. Although this is a conscious choice of the authors, as the primary interest of this study concerns the performance of various types of flood insurance arrangements under different climatic- and socio-economic scenarios, the inclusion of some additional behavioral theories can also provide interesting insights. For example, an agent-based study found that interaction between individuals, insurance markets, and the environment significantly affects future flood risk due to additional adaptive behavior by heterogenous boundedly rational individuals [67]. Similarly, an agent-based approach could be applied to model intra-household behavior concerning insurance purchase and the decision to apply risk-reduction measures. Existing research suggests that this type of complexity of within-household decision making concerning consumption exists [68]. However, to our knowledge no previous studies exist that relate this type of decision making to flood insurance purchase decisions or adaptation to natural hazards. A downside of modelling more complex household behavior on this level, especially using agent-based models, is that it is computationally too demanding on a large scale, such as EU-level.

The unaffordability of flood insurance, as well as the penetration rate, are found to improve when introducing various elements of the Public-Private-Partnership (PPP). Paudel et al. [69] explains how this type of insurance system can lower premiums by shifting the extreme risk (top 15% in our model) to a public reinsurer that does not charge a profit-loading factor for the transferred risk. This is made possible by its "*non-profit status, low administrative costs and lack of risk load*" [70]. An example of a public reinsurer, although in a slightly different form as described earlier, is Flood Re in the UK, which was able to reduce premiums up to 50% for households with a prior flood claim since 2016 [71]. Our study found the highest reduction in unaffordability of premiums by incorporating an element of risk-sharing in the form of a premium-cap. In principle, public reinsurance can be combined with a premium-cap in a PPP flood insurance arrangement in order to limit problems with unaffordability of premiums.

In addition to introducing a public reinsurer or premium cap, we include insurance schemes with purchase requirements. This includes a semi-voluntary scheme, where insurance uptake is required for mortgages, such as in Sweden, or included in home and real estate insurance, such as in Finland. As a result of this measure, both of these countries are able to maintain penetration rates above 80% [72,73]. Besides conditional purchase requirements we include an insurance scheme that is mandatory for all, as can be found in France, Spain, Belgium, and Romania. All of these countries are able to maintain very high penetration rates, except for Romania, where the enforcement of the insurance purchase mandate is lacking [72]. These type of insurance policies limit problems with adverse selection compared to a voluntary system, as most households in lower risk areas cannot choose to opt out of insurance. As a result of this, insurers are better able to spread risk, which can enhance insurer solvency and potentially reduce premiums for households [69]. Moreover, high penetration rates mean that most households are financially protected against flood damage. This limits the impact on private savings after flood events, or the burden for the public budget when the government offers disaster relief to uninsured households, as was, for instance, required after the 2013 floods in Germany [74].

A relevant implication of introducing insurance purchase requirements is that the total value of unaffordability must be compensated, as forcing households into poverty through mandating coverage may have large socio-economic consequences. An existing measure to tackle the problem of unaffordability is to provide subsidized insurance to low income households, or households residing in high-risk areas [75]. A subsidy can be either provided directly to households, or to insurers who can then charge premiums below the expected value of loss. A downside of these measures is that households in high-risk areas receive less incentive to mitigate or adapt to flood risk, which can have a negative welfare consequences, as more funding will be required for subsidies due to rising flood risk. Kousky and Kunreuther [7] propose a voucher mechanism as an alternative, in which means-tested vouchers are coupled with low-cost loans meant for investment in risk-reducing measures. Alternatively, governments can subsidize risk-reduction measures for low-income households. This, in turn, will lower flood risk, which reduces the cost of insurance and limits unaffordability issues. Instead of a lump-sum transfer to cover unaffordability, this measure reduces overall flood risk and, therefore, is more efficient.

Alleviating premium costs for households in high-risk areas is justified from a solidarity perspective. However, it should be considered that it can stimulate economic development and increase exposure to flooding in high-risk areas [76]. Therefore, premium reductions should be accompanied by incentives to mitigate risk, such as premium discounts when risk-reduction measures are implemented. An example of this type of policy is the community rating system implemented by the National Flood Insurance Program in the US, where households can receive a premium discount based on their community's effort to reduce risk [77]. Moreover, individual households can receive additional premium discounts if people elevate the lowest floor of their home, which is a cost-effective way for limiting flood risk especially for newly constructed homes [78]. A disadvantage of stimulating risk reduction with premium discounts is that it involves administrative costs for the insurance company to determine whether the risk reduction measures are in place. In

practice, this can be overcome through external building certification schemes, such as elevation certificates issued in the US.

Relocation to areas that are less prone to flooding can be another way to cope with increasing flood risk and higher flood insurance premiums, besides structural measures that reduce flood risk. Although large-scale migration in response to natural disasters has not been observed in developed countries [79], some studies suggest this as a viable option to enhance future societal resilience to flooding [80]. Risk-based insurance premiums can stimulate this type of adaptation when these rise rapidly in high-risk areas. However, insurance policies should beware that this does not disproportionally cause problems for low-income groups, as relocating may be infeasible or too expensive for these groups. If relocation of households from high- to low flood-risk areas will happen on a larger scale towards 2080, then the level of unaffordability of premiums simulated in this study may be overestimated.

The mentioned policy changes considered several potential insurance policy strategies and how they impact unaffordability of premiums and uptake of coverage. Importantly, what we did not address in this study is how an improvement in flood protection standards can impact these outcomes. In the flood risk module, it was assumed that protection standards remain constant in the case flood hazard increases, meaning that governments invest in maintaining protection infrastructure to grow along with the flood hazard. However, additional public spending can be directed to enhance flood protection standards, which can significantly reduce regional flood risk, and therefore reduce insurance premiums without additional changes in insurance strategies. However, relying on this measure may enlarge regional inequality in flood insurance affordability and availability, as enhancing protection standards in the face of increasing flood hazard can be costly, and may be unaffordable for lower income countries.

Finally, in this study, we considered which European regions are projected to experience growing unaffordability of premiums and declining uptake of insurance to such an extent that it obstructs the functioning of insurance markets, leaving households to finance flood damage by different means. However, as this analysis focusses on high-risk areas, a perceived regional tipping-point may not lead to a collapse of private flood insurance on a national scale. Instead, the collapse of flood insurance markets was approached from the consumer side, where unwillingness to pay for rapidly increasing premiums by households causes the insurance system to no longer meet its objective, which is to provide financial security for citizens against the growing threat of flooding. Furthermore, tipping-point regions are identified using criteria selected by the authors based on projected insurance penetration rates. The criteria used here define the occurrence of a tipping-point when the penetration rate declines from 20% in 2010 to less than 5% in 2080, which is a selected point where insurance fails to provide coverage for a significant share of households. However, this definition is not absolute, and changing these criteria also affects the number of tipping-point regions.

## 5. Conclusions

This study showed the possibility of a climate-induced socio-economic tipping-point for flood insurance in Europe, where rising unaffordability and declining uptake of insurance under climate change obstructs the functioning of insurance markets in high-risk areas. As a result, regional inequalities arise in the ability to use flood insurance as an instrument for adapting to increasing flood risk. We observe progressively rising flood insurance premiums over time from the climate change scenarios RCP2.6 to RCP8.5, for countries that maintain risk-based insurance premiums. This causes an increase of unaffordability of insurance, the extent of which depends on the projected socio-economic development in that period. Such unaffordability problems occur particularly in regions with below average income per capita. When insurance uptake is voluntary, the increase of premiums combined with underestimated household risk perceptions causes a decline of insurance demand. This process causes flood insurance demand to almost disappear for several regions in our projections. The number of European regions where this occurs is highest under the RCP8.5-SSP3 scenario, which represents a continued rise in greenhouse gas emissions combined with regional economic inequalities.

This development constitutes a socio-economic tipping-point as the collapse of private flood insurance calls for a shift of flood damage compensation from pre-funded, formal insurance, towards less formal means of financing, such as ex post government compensation or self-insurance. However, *ex post* government compensation is a contingent liability, which, unless made explicit, is inherently uncertain for both the government's budget as well as the household. Therefore, we argue in favor of formal insurance, which can be sustained in the face of increasing flood risk by maintaining premiums with limited risk-reflection, as this encourages risk-mitigation while limiting unaffordability of premiums. Furthermore, coverage requirements should increase the covered assets and enhance the ability for insurers to pool the risk. Finally, a government reinsurer can reduce premiums as it is risk neutral and does not seek profit.

We recommend future research to focus on developing a more specific strategy for implementing a voucher mechanism that covers the remaining unaffordability of insurance. This system should alleviate the share of the insurance premium that is considered unaffordable, while giving an incentive for households to implement risk-reduction measures and reduce the premium. We also recognize the importance of investigating the impacts of uncovered flood risk on household and government budgets in future research. A more extensive analysis of household behavior and decision making under scenarios of flood risk is another topic that deserves attention in future research. Research can, for example, focus on the performance of insurance markets or the extend of household-level adaptation to flood risk under multiple theories of individual decision-making, such as decision rules that account for myopia or are based on heuristics.

**Author Contributions:** Conceptualization, M.T., W.J.W.B., and P.H.; data curation, T.T.; formal analysis, M.T.; funding acquisition, W.J.W.B. and J.C.J.H.A.; investigation, M.T.; methodology, M.T. and P.H.; project administration, W.J.W.B. and J.C.J.H.A.; resources, P.H.; software, T.H. and P.H.; supervision, W.J.W.B., T.H., and J.C.J.H.A.; validation, W.J.W.B.; visualization, M.T. and T.T.; writing—original draft, M.T.; writing—review and editing, W.J.W.B., T.H., P.H., T.T., and J.C.J.H.A. All authors have read and agreed to the published version of the manuscript.

**Funding:** This work was funded by the Netherlands Organization for Scientific Research (NWO) and the EU Horizon 2020-project COACCH. NWO: VICI grant no. 453.14.006 & VIDI grant no. 452.14.005; COACCH grant no. 776479.

**Conflicts of Interest:** The authors declare no conflict of interest.

## Appendix A. Premiums in the Solidarity and Public-Private Market Structures

*Appendix A.1. Solidarity Market (M1)*

This market equally spreads the national annual flood loss across the population, resulting in a high degree of risk sharing that limits unaffordability issues. Equation (A1) expresses how the premium is modelled in the DIFI framework. The numerator equals the portion of the flood risk that is transferred to the insurer, which is calculated as the expected value of the flood damage ($L_{c,t}$), on a country level $c$ for time $t$, from which the total deductible value is subtracted. $D_{c,t}(p)$ is the total value of the deductible (set at 15% of the claimed loss) that must be paid by households in country $c$, at time $t$, with probability $p$. The final premium is a flat-rate tariff for all households in a country $N_{c,t}$ both inside and outside of the high-risk area.

$$\bar{\pi}_{c,t} = \frac{E\left(L_{c,t}(p) - D_{c,t}(p)\right)}{N_{c,t}} \tag{A1}$$

*Appendix A.2. Public-Private Market (PPP) (M6)*

The premium in this market is shown in Equation (A2), which differs from the solidarity market in three main ways.

First, the premium is determined at the NUTS2-level, hence the subscript *j* is introduced, which represents that geographic level. This implies that $\bar{L}_{j,t}$ and $\bar{D}_{j,t}$ are the modelled flood damage and deductible per high-risk household on this geographical level.

Second, the final premium charged is the risk-based premium, unless it exceeds a specific threshold, which is set to create a certain cross-subsidization between high- and low-risk households. This threshold, or premium cap, is set at 1.8% of median income in the region, following Hudson et al. [26].

Third, specific cost loading factors are introduced for the proportion of risk in the PPP that is covered by private primary insurance, or public reinsurance. $\dot{\lambda}_{c,t}$ is the cost-loading factor for the primary insurer; $\ddot{\lambda}_{c,t}$ is the cost-loading factor of the public reinsurer operating on a national level; the superscript *RR* indicates the risk transferred from the insurer to the reinsurer; *r*, is the risk aversion coefficient of the private insurers; $\sigma_{0<a<99.8}$, is the volatility of flood damage within a quantile range that is considered insurable [25].

The loading factors for the primary insurer and public reinsurer are determined as follows. For the private primary insurers Bertrand competition is assumed, which means that no company can charge a profit-loading factor. Concerning the public reinsurer, the government is assumed to have no interest in charging a profit-loading factor. The cost-loading factors are determined on a country-level based on OECD insurance statistics of costs of non-life insurance policies, as is described in detail in Hudson et al. [26].

$$\bar{\pi}_{j,t} = min \begin{cases} \left(1 + \dot{\lambda}_{c,t}\right)\left(E\left(\bar{L}_{j,t}(p) - \bar{D}_{j,t}(p)\right) + r * \sigma_{0<a<99.8}\right) \\ +\left(1 + \ddot{\lambda}_{c,t}\right)\left(E\left(\overline{L_{j,t}^{RR}}(p) - \overline{D_{j,t}^{RR}}(p)\right) + r * \sigma_{0<a<99.8}^{RR}\right) \\ \left(1 + \ddot{\lambda}_{c,t}\right)CAP_{j,t} \end{cases} \quad (A2)$$

The final premiums faced by individuals $(\pi_{i,j,t},)$ is determined as the baseline average risk per household for a certain NUTS2 region, which is reduced by the premium discount (ER_DRR), expressed as a fraction of the total premium. The premium discount resulting from the risk reduction measures implemented by policyholders is determined in the household behavior module [26].

$$\pi_{i,j,t} = (1 - ER_{DRR})\bar{\pi}_{j,t,s} \quad (A3)$$

Additionally, the premiums required to maintain a total flow of premiums equal to the expected loss is achieved by placing a small surcharge on the premiums of lower risk policyholders.

## Appendix B. Results of Unaffordability and Market Penetration under Scenarios: RCP2.6-SSP2; RCP6.0-SSP2; RCP8.5-SSP5

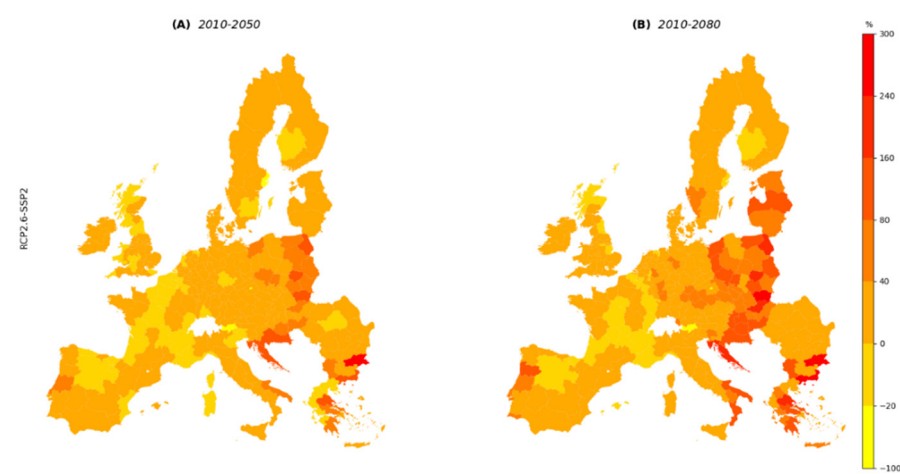

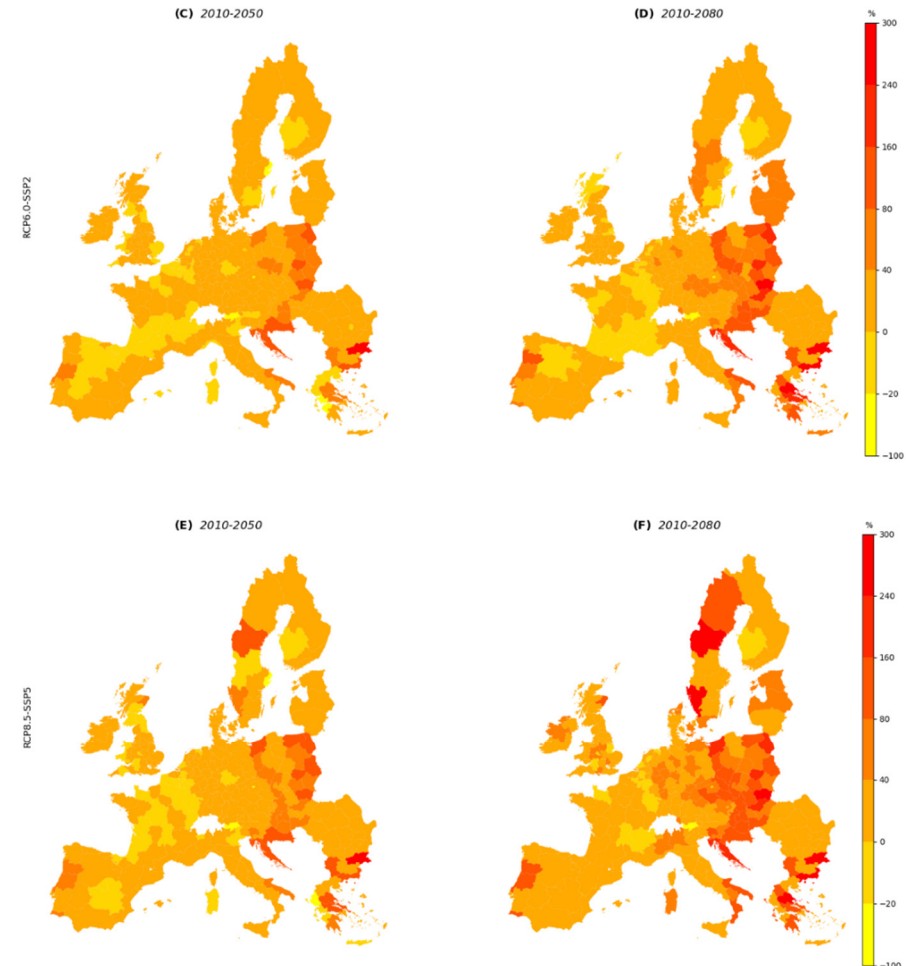

**Figure A1.** The percentage change of unaffordability under status-quo insurance arrangements for households in high-risk areas under the RCP2.6-SSP2 (panels **A** and **B**), RCP6.0-SSP2 (panels **C** and **D**), and RCP8.5-SSP5 (panels **E** and **F**) scenarios of climate and socio-economic change for the periods 2010–2050 (left) and 2010–2080 (right). Unaffordability is measured as the percentage of the population in high-risk areas that cannot afford the insurance premium.

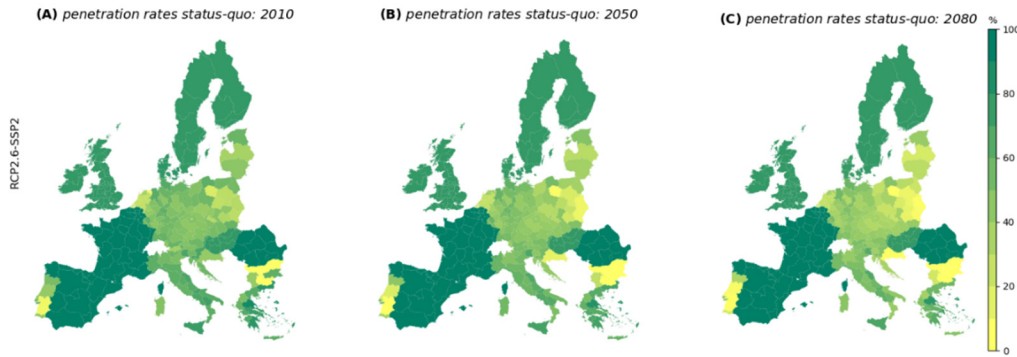

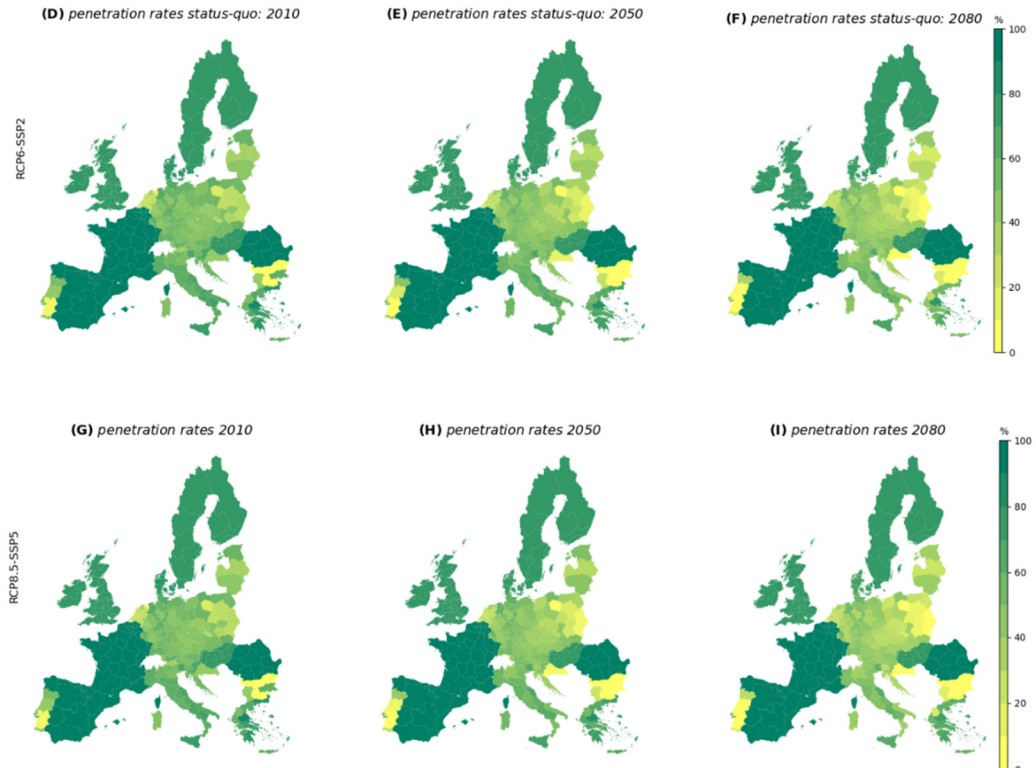

**Figure A2.** Flood insurance penetration rates under status-quo insurance arrangements for households in high-risk areas under the RCP2.6-SSP2 (panels **A–C**), RCP6.0-SSP2 (panels **D–F**), and RCP8.5-SSP5 (panels **G–I**) scenarios of climate and socio-economic change for the periods 2010 (left), 2050 (middle), and 2080 (right).

## Appendix C. Expected Annual Damage (EAD) in the Baseline and for Future Projections under Multiple Scenarios

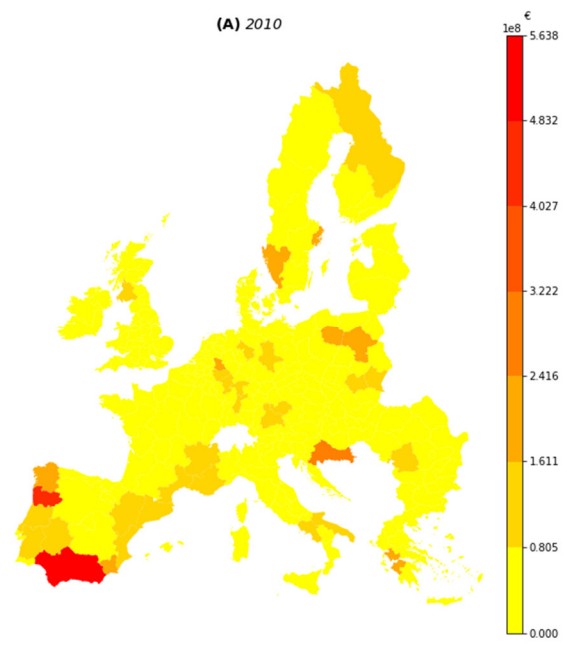

**Figure A3.** Expected annual damage for NUTS2 regions in 2010.

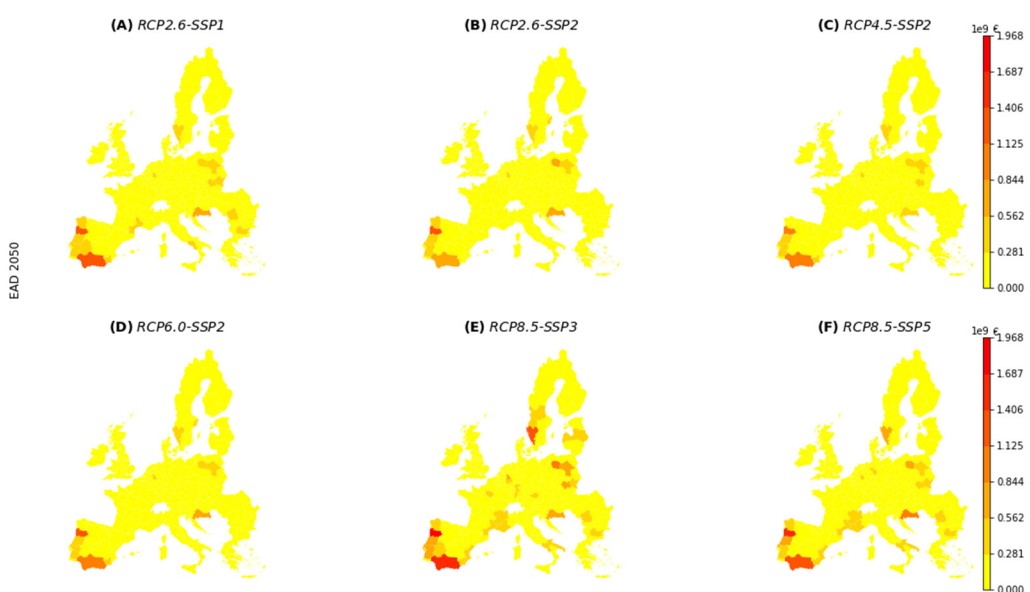

**Figure A4.** Expected annual damage in 2050 under RCP2.6-SSP1 (panel **A**); RCP2.6-SSP2 (panel **B**); RCP4.5-SSP2 (panel **C**); RCP6.0-SSP2 (panel **D**); RCP8.5-SSP3 (panel **E**); RCP8.5-SSP5 (panel **F**).

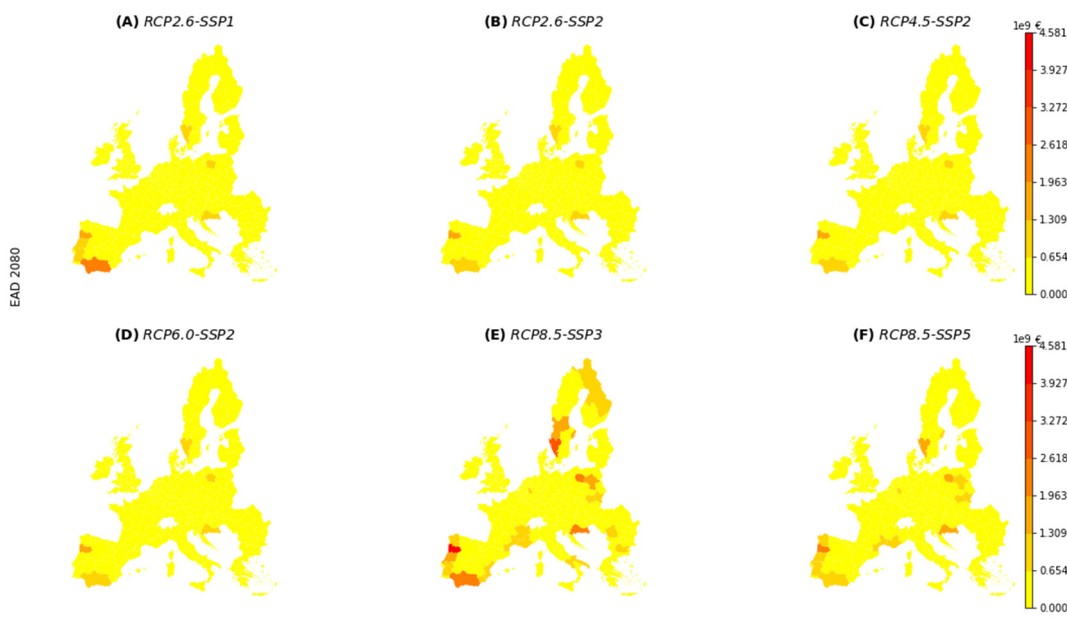

**Figure A5.** Expected annual damage in 2080 under RCP2.6-SSP1 (panel **A**); RCP2.6-SSP2 (panel **B**); RCP4.5-SSP2 (panel **C**); RCP6.0-SSP2 (panel **D**); RCP8.5-SSP3 (panel **E**); RCP8.5-SSP5 (panel **F**).

## Appendix D. Categorization of Current Insurance Market Structures in the EU

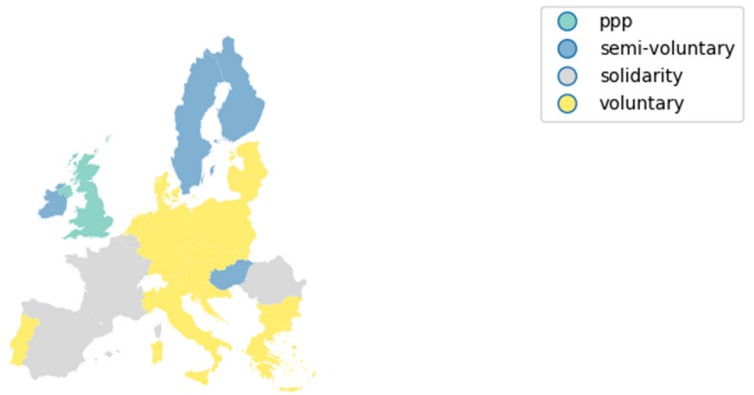

**Figure A6.** Categorization of current insurance market structures in the EU (see Table 1). Source: Tesselaar et al. [81].

## Appendix E. Flood Insurance Premiums in the Baseline and Future Projections Using Multiple Scenarios

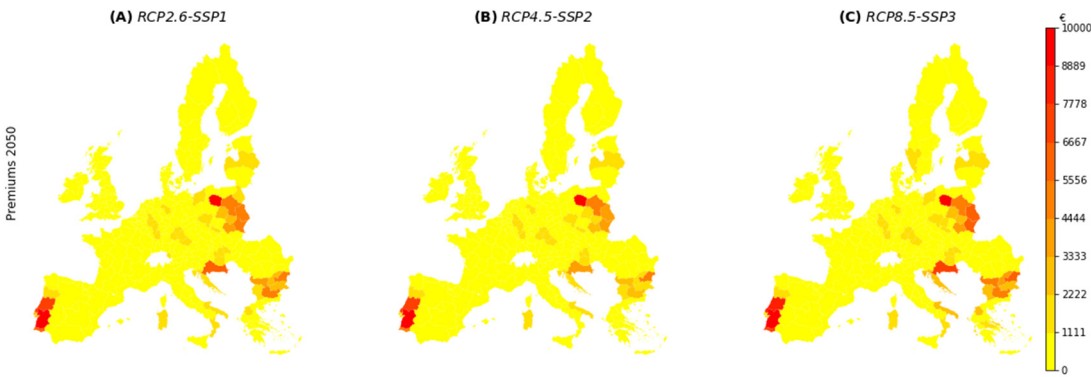

**Figure A7.** Flood insurance premiums in 2050 on NUTS2-level, under current national insurance systems, shown for RCP2.6-SSP1 (panel **A**); RCP4.5-SSP2 (panel **B**); RCP8.5-SSP3 (panel **C**).

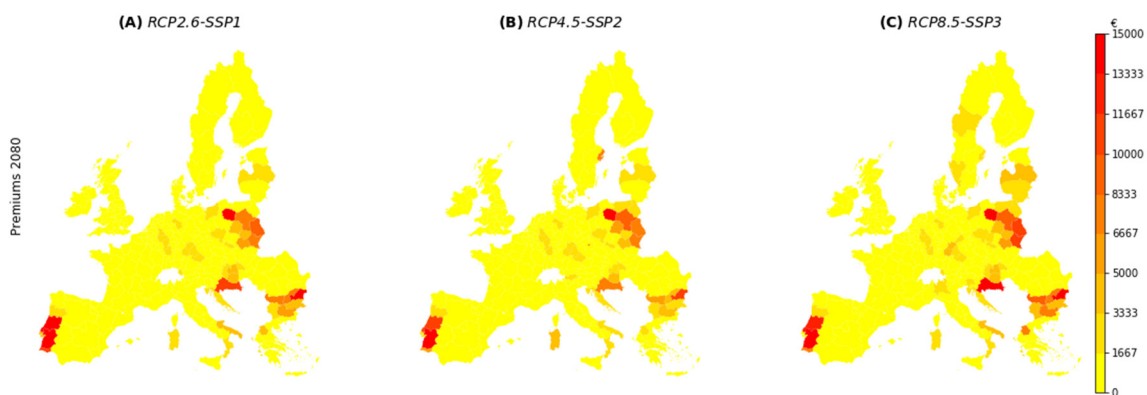

**Figure A8.** Flood insurance premiums in 2080 on NUTS2-level, under current national insurance systems, shown for RCP2.6-SSP1 (panel **A**); RCP4.5-SSP2 (panel **B**); RCP8.5-SSP3 (panel **C**).

## Appendix F. Map of EU Countries and the UK

**Figure A9.** Map of EU countries. The United Kingdom is outside of the EU-region since 1 February 2020, but is still included in the DIFI model. *Source Europa.eu.* Retrieved from: https://europa.eu/european-union/about-eu/easy-to-read_en (accessed 5 October 2020).

**Appendix G. List of Variables and Their Definitions**

| | |
|---|---|
| $\pi_{i,j,t}$ | - Final premium faced by individual household $i$ in region $j$ at time $t$. |
| $p$ | - Flood occurrence probability. |
| $\bar{L}_{j,t}$ | - Expected flood damage. |
| $\bar{D}_{j,t}$ | - Deductible. |
| $\dot{\lambda}_{c,t}$ | - Cost-loading factor for the primary insurer. |
| $r$ | - Risk aversion coefficient of the primary insurers and reinsurers. |
| $\sigma_{0<a<99.8}$ | - The volatility of flood damage within a quantile range that is considered insurable. |
| $\ddot{\lambda}_{c,t}$ | - Cost-loading factor for private reinsurers. |
| $\overline{L_{j,t}^{RR}}$ | - Expected flood damage that is transferred from insurers to reinsurers. |
| $D_{j,t}^{RR}$ | - Deductible set by reinsurers for primary insurers. |
| $ER_{DRR}$ | - Premium discount that is determined by the implementation of risk reduction measures. |
| $\widetilde{PS}_{i,j}$ | - Maximum flood occurrence probability considered by households. This is determined by $\widetilde{PS}_{i,j} = \vartheta_{i,j} PS_j$ where $PS_j$ is the regional flood protection standard, and $\vartheta_{i,j}$ is a distribution of subjective flood occurrence probabilities. |
| $W_{i,j,t}$ | - Wealth of individual $i$ in region $j$ at time $t$. |
| $\gamma_{i,j}$ | - Flood impact misperception parameter. |

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
