# Peer review of "Regional Inequalities in Flood Insurance Affordability and Uptake under Climate Change"

_sustainability, doi:10.3390/su12208734_

Round 1
Reviewer 1 Report
Dear Authors,
I have read your manuscript carefully. I found it extremely precise and acute both in the description of the methodology and in the explanation of the results. the form and the substance coincide in an excellent result. I have no observations or comments to add.
Sincerely
Author Response
We thank the reviewer for taking the time to review the manuscript, and we are grateful for the kind words the reviewer states regarding our study.
Reviewer 2 Report
The paper is a technically expansive attempt to grapple with the important questions of whether climate change might lead to the unaffordability of flood insurance in different countries of Europe. In order to do so, the paper combines the output from four separate models – a riverine flood model, a flood impact model, a model of the insurance sector, and a household behaviour model. Many aspects of these models stem from previous research, most notably the DIFI model. The paper argues that under certain climactic prediction, various European countries could experience “a climate induced socio-economic tipping point” where flood insurance becomes unaffordable. The paper looks at the climate projections for two years, 2050 and 2080, and shows that under different climactic scenarios, a number of regions in Europe reach a tipping point in the unaffordability of flood insurance. The paper then goes on to make some suggestions about the type of insurance market best suited for these scenarios.
While I applaud the choice of topic and I think this is an important piece of research, I think the authors do not allow the reader to adequately follow the different steps in their overall models. I also have very strong reservations about aspects of the model, especially the household behaviour model. I think the authors fail to adequately discuss and highlight many of the problematic assumptions which are made. My overall suggestion would be to radically improve the household behaviour model as well as introduce an entire new section (which I would place before the results section) which discusses the limitations and central assumptions. I would also prefer if the authors strengthened the discussion of the economic aspects of the model.
The paper provides an excellent introduction [lines 39-125] which nicely lays out the main considerations to take into account when considering the affordability of flood insurance.
- Actual Damage of flood risk model never specified
This comment will re-occur in a number of sections. The problems I have with the paper is even though the paper estimates the affordability of floods, nowhere are there any actual numerical estimates to be found. I would like to see how current and historical flood damages compare to expected annual damages (EAD line 145) under the different climate scenarios. A simple table for different regions which compares historical and expected damage would suffice. Otherwise, the reader has no sense of whether the paper’s various assumptions even make sense.
- Very weak consumer behaviour model (2.3. line 277 onwards) (side note: sometimes in the paper this model is called consumer model, other times household model).
- Expected utility not acceptable theory for research question.
Although there is a long history of scholarship that uses subjective expected utility theory to study individual decision-making, there is a nearly equally long history of scholarship that highlights the limitations. Personally, I strongly believe that modelling needs to go beyond such a simple model in the year 2020. In fact, the authors even unwittingly admit the inapplicability of subjective expected utility theory. In line 73, the authors state that “individuals tend to underestimate risk of events that occur with a low probability”. This is an impossibility in a subjective expected utility framework as individuals are assumed to always attach the right probabilities to events. The fact that the authors are aware of the limitations of expected utility theory clearly demonstrates that a more sophisticated model is needed here. The model should take into account asymmetric information, imperfect information, and behavioural limitations on the side of agents. One suggestion would of proceeding is to combine the simple model with other types of economic agents (myopic, attribution bias for example and some others) and to compare the outcomes between four of five different kinds of agents. Or to model the agents in a way that is simply more sophisticated and convincing. I would also be very interested to see how whether simply heuristics (see work of Gerd Gigerenzer for example) would provide other outcomes. I find this particularly important as the authors have gone to the length to differentiate different insurance markets. In the same way as they do not assume perfect competition, I think the consumer model deserves to be at least as nuanced as the insurance sector model.
- Tipping points never specified – should be done in consumer model
Even though the paper’s central argument is about tipping points of the unaffordability of insurance, nowhere in the paper do we have any numerical estimates of what this unaffordability amount to (see point 1). Rather than seeing a standard textbook explanation of how expected utility theory works in relation to a budget constraint, I think this section should strive to derive numerical monetary values. I would like to see what in monetary terms what flood insurance currently cost per household in different regions and what the projection is for how much this would have to cost in 2050 and 2080 under different insurance schemes and climactic scenarios respectively. Also I would personally prefer to entire discussion to take place in terms of affordability. I.e. rather than looking at two fairly arbitrary years and show the proportion of unafforability by region, I would prefer a full times series showing proportion of income required for flood insurance. So that we see the evolution over time from today to 2080 and the authors can then comment on when they think the tipping point is reached, by region.
- Conflation of individual and household.
I know that this is frequently done, but I do not agree to treat a household like a single individual. There are different individuals in a household with separate indifference curves and many funny things can happen when they have to agree on what to do (think Arrow’s impossibility theorem). So maybe some commentary on what might happen when members of the household disagree might be necessary
- No discussion of exit (moving)
I also find it perplexing that the entire paper does not once mention one obvious solution to flood unaffordability – that is moving. If insurance becomes too expensive, then people have to move. Maybe the authors disagree with this point but in any case it should be discussed and how it relates to the arguments and findings of the paper.
- Confusion between risk and uncertainty and different conceptions of probability
- The paper also suffers from the fact there is a continuous confusion between risk and uncertainty. I am using the terms here as is standard in economics, where risk is quantifiable and uncertainty is unquantifiable (see seminal discussion by Frank Knight or recent book by Kay and King (2020) on radical uncertainty. At numerous times at the paper, the authors call something risk when it is in fact an uncertainty. Climate events of the future do not have a known exact probability distribution so they should not be treated as if they do. This is especially relevant in the consumer model and insurance sector mode.
- Even worse, the paper uses different conceptions of probability. In the insurance sector model and flood model, the losses are presumably based on objective probabilities, i.e. probability of events (not subjective probability of one insurance company). Yet, the expected utility theory model uses subjective (Bayesian probabilities) which have nothing to do with the probabilities used in the other two models. So the P of expected losses for the same event will be different for consumers as for insurance companies (which again introduces some of the discussion of point 2). The difference between these probability assessments needs to be discussed much more explicitly as well as the complications that derive therefrom.
- No mention of economic growth and income growth model used
This is a huge oversight. I cannot find any explanation in the paper about how consumer incomes have been modelled into 2050 and 2080. In fact, it is even not clear to me whether this has been done. There is a mention in line 375 of “below average projected income growth” but the reader is not told what this growth is, where it comes from etc. In particular, I am worried that the changes in affordability is not driven by climactic changes but simply reflects relative economic growth differentials. Again, a sensitivity analysis that differentiates different rates of economic growth in different regions with different climactic scenarios would be needed here to allow readers to differentiate what is driving unaffordability.
- GDP growth is not equal to household income growth
Just as a note of caution, I want the emphasise the authors should not simply use GDP growth figures as they are interested in lower income households and there is no guarantee that this income grows at same rate as GDP. Case in point is that real US GDP since 1965 has risen by a factor of about 5 (from about 4 trillion to about 20 trillion), whilst real wages have basically been stagnant (risen by a factor of about 0.1). Clearly, the source of growth matters and as does income distribution. Again, all of this should be discussed. If the paper is truly interested in whether flood insurance becomes unaffordable for poorer households, there needs to be an explicit discussion of income distributions used in each country (again, is unaffordability simply being driven by larger income inequalities?)
Reviewer 3 Report
This paper is impressive in many respects. It is a good example of modeling a simulation, using various parameters to estimate possible outcomes.
My preference would be that the authors said more about the problems of moral hazard and adverse selection in their discussion of voluntary flood insurance. I think that is crucial to understanding the market for this kind of insurance and the social costs of having this market fail. I think the authors might also say a bit more about the national policies with respect to flood insurance. They place different countries into different categories of types of flood insurance policies, but i think they leave out a great deal of institutional detail. I am a bit unclear about the use of flood insurance by non-household property owners. Most of the discussion treats households as the primary consumers, but i doubt that all of the relevant owners are households. I expect many are businesses and I suspect that many are government entities. In Europe, much of the housing is not owner-occupied, and much of it is publicly owned, so the number of households involved many not be as great as implied. In any case, there will be enormous variation in the number of property owners in affected areas across countries.
With respect to the discussion of the significance of the results, it is probably important to cover the implications of different forms of subsidy. I think it likely that voluntary flood insurance is unsustainable in high risk areas. There will be huge problems of adverse selection unless everyone or almost everyone has insurance. The question then becomes affordability. The authors suggest a premium cap along with public reinsurance. That is one possibility. Another would be the means-tested voucher that Kousky and Kunreuther suggest. Another would be a subsidy that goes directly to the insurance companies based upon the estimated liability not covered by premiums. I do think that section should be expanded.
I have one stylistic complaint. The paper is very much filled with acronyms. Some are unavoidable, but some are unnecessary. For example, in labeling scenarios, is it really necessary to refer to RCP8-SSP3, et al.? please come up with some label for that scenario that conveys something of its nature without using a bizarre combination of characters.
Reviewer 4 Report
The study focuses on how future flood risk will affect vulnerable insurance policy holders and is important and timely. The study itself is interesting and has merit. However, the article needs improvement to fit within the aims and scope of the Sustainability journal. Sustainability is a broad reaching journal both in article subjects it accepts but the locations of the studies and its readers.
This is where the article’s primary issues start. Throughout the article, it appears the authors assume the readers have more knowledge than they may have.
For example, an explanation of the NUTS2 regions is relegated to a brief footnote. But these regions are a major unit of analysis for the entire study. The authors should spend more time explaining the NUTS2 system.
I also spent considerable time trying to understand variables within each equation in the methods section. What each variable means, where they are calculated, and if they come from a previous equation. A better organization of the variables within each equation would be helpful. Perhaps a bulleted list of each variable containing a brief description would likely remove considerable work for the reader to understand how the equations work.
Throughout the article the authors identify regions of Europe without previously providing any context for the different countries. If this journal was primarily European this would be fine, however, since it is not, some context is necessary. This could be as simple as labeling some regions on the maps that they identify in the results.
I would also like to see Table 1 incorporate a map of the “structure groups”. This would help the reader to understand how changes in risk in subsequent maps are influenced by the structure group.
Table 2 could also use more attention. Currently, it is not very organized. The number of regions and number of NUTS2 regions is redundant across the table. The countries are not in any obvious order, but some are removed from as the reader moves right. If the table was organized by those that remain constant at the top and those that are removed as the reader moves right, Czech and Poland could be placed at the bottom. This would allow a column for the regions and NUTS2, removing redundant information and allow the new columns headers making the table capable of standing alone without the description.
Finally the use of footnotes is inconsistent through out the article. For example
Footnotes 1 and 4 could be citations within the main document.
Footnote 7 could be fleshed out within the actual paragraph and provide more context to the reader.
Footnote 10 merits further discussion in either the results or as part of the discussion section.
Round 2
Reviewer 2 Report
I would like to congratulate and thank the authors for the thoroughness with which they have engaged with my comments. I am very impressed not only by the speed but by what is clearly a heartfelt attempt to follow my suggestions, as far as this was possible. As far as I am concerned, I am basically satisfied, except for some tiny stylistic changes and one more series of graphs if possible.
- For the graphs page 27 and 28, could you please use a notation of billions and trillions. I would find that just easier to read than the current notions (5.4e^9).
- For the graphs on page 29, please use a comma to separate after each 1000, so 15499 is 15,499.
- Could you please add one more series of graphs into the appendix? I would very much like to see the same graphs as page 27 and page 28, where the expected annual damage is shown as a percentage of GDP in the same year, both in baseline and predictions.